# Investigation of Microstructures and Mechanical Properties of SiC/AA2024 Nanocomposites Processed by Powder Metallurgy and T6 Heat Treatment

**DOI:** 10.3390/ma15103547

**Published:** 2022-05-16

**Authors:** Dikunqi Mu, Zhen Zhang, Jiamiao Liang, Jun Wang, Deliang Zhang

**Affiliations:** 1Shanghai Key Lab of Advanced High-Temperature Materials, Precision Forming and State Key Lab of Metal Matrix Composites, Shanghai Jiao Tong University, Shanghai 200240, China; onlymed@163.com (D.M.); zhangzhen1215@sjtu.edu.cn (Z.Z.); junwang@sjtu.edu.cn (J.W.); 2State Key Laboratory of Rolling and Automation, Northeastern University, Shenyang 110819, China

**Keywords:** aluminum alloy matrix composites, SiC nanoparticles, microstructure, mechanical properties

## Abstract

SiC/AA2024 nanocomposites with 1 and 5 vol.% SiC nanoparticles have been prepared by a powder metallurgy route involving high-energy ball-milling (HEBM), spark plasma sintering (SPS), and hot extrusion. The microstructures and mechanical properties of the nanocomposite samples before and after T6 heat treatment were investigated. The samples exhibited a bimodal microstructure with SiC nanoparticles being dispersed in it. With increasing the SiC nanoparticle content from 1 to 5 vol.%, the yield strength (YS) and ultimate tensile strength (UTS) increased and the elongation to fracture (El) slightly decreased. After T6 heat treatment, a simultaneous improvement of the strength and ductility was observed, with the YS, UTS, and El increasing from 413 MPa, 501 MPa, and 5.4% to 496 MPa, 572 MPa, and 6.7%, respectively, in the 1 vol.%SiC/AA2024 nanocomposite sample. Analysis of the deformation behavior shows that this improvement is likely caused by the increased density of geometrically necessary dislocations (GNDs) resulting from the bimodal microstructure. The dispersed intragranular Sʹ precipitates generated by the T6 heat treatment also make a contribution to the increase of strength and ductility by accumulating dislocations. It is feasible to realize simultaneous improvement of strength and ductility in the SiC/AA2024 nanocomposites via powder metallurgy and subsequent heat treatment.

## 1. Introduction

Aluminum alloy matrix composites (AMCs) reinforced with nanoparticles have been attracting great attention [1,2] as potential advanced materials for applications in many fields, especially in aviation, space, and automotive industries, because of their high specific strength, high elastic modules, and excellent wear resistance. It is well-recognized that the mechanical properties of AMCs are affected by the addition of ceramic nanoparticles [3]. However, it is still a challenge to successfully introduce dispersed nanoparticles into the matrix. Yao et al. [4] found that with increasing SiC content from 1 to 10 vol.%, the agglomeration of SiC nanoparticles and porosity in the SiC/AA6063 nanocomposites produced by powder metallurgy caused fracture elongation to decrease from 10% to 2.3%. Park et al. [5] have prepared 0.1–0.4 wt.% carbon nanotubes (CNTs)/Al composites by sintering and the melt-blending method. They found that the presence of CNT clusters and Al_4_C_3_ nanoparticles led to a premature fracture with raising the content of CNTs, and the best mechanical properties were achieved with the addition of 0.2 wt.% CNTs. Canakci et al. [6] found that the hot-pressed density (final density) of the composites decreased with increasing amounts of SiC content, while an increase in the weight percentage of the SiC particles (from 0 to 10 wt.%) in the AA7075/Al–SiC composites improved the hardness of the recycled composites due to the high hardness of the SiC particles.

Nanocomposites often possess improved tensile properties but poor work-hardening ability due to the significant refinement of matrix grain sizes [4,7]. Among the material processing methods, powder metallurgy (PM) has shown great advantages in dispersing nano-reinforcements uniformly and achieving good properties [8,9]. A novel PM route was proposed by Fan et al. [10] to fabricate strong and ductile CNT/Al composites reinforced with high-content CNTs. As a result, 3 vol.% CNT/Al composites exhibited good mechanical properties, with tensile strength enhanced by 65.7% and modulus increased by 28.9% as compared to the unreinforced Al matrix, while keeping a fine ductility of 8.8%. Roshan et al. [11] studied the microstructure and tensile properties of nanocomposite AA2014–4 wt.% Al_2_O_3_ produced from machining chips using PM. Their results showed that Al_2_O_3_ nanoparticles were very well-introduced and dispersed in the AA2014 matrix after the milling, but they reported an intergranular fracture of extruded particles that was accelerated by weak grain boundaries (GBs). Using semi-solid powder rolling, 10% B_4_C–AA2024 composite strips were prepared by Mo et al. [12]. The study revealed that the optimal fraction of liquid phase ranged from 38% to 60% and the B_4_C particles could restrain the generation of microcracks during the solidification process. One of the main advantages of the powder metallurgy method is the low process temperatures without the absence of a chemical reaction between the matrix and reinforcement. In addition, a relatively higher fraction of reinforcement particles can be introduced in the composite when compared to the traditional processes [13].

Some methods have been explored to improve the ductility of nanocomposite materials, including introducing intragranular nanoparticles [14,15] by precipitation, as well as developing a heterostructure [16,17]. However, it is not easy to develop an improved procedure for simultaneously enhancing the strength and ductility. Recently, heterostructured metallic materials, including heterogeneous laminate structure [18], gradient structure [19], and bimodal structure [20,21], have attracted extensive attention from the materials research community. These diverse structures have a common feature: all of them consist of both soft domains and hard domains with dramatically different flow stresses. During tensile deformation, geometrically necessary dislocations (GNDs) are generated due to the differentiation in deformation of domains of different hardness, and this in turn produces back-stress-induced hardening and helps with retaining tensile ductility [22].

In this study, we fabricated SiC nanoparticle-reinforced AA2024 matrix nanocomposites by a powder metallurgy route, and investigated their microstructures and tensile properties. The purpose of this study is to gain a better understanding of the effect of SiC nanoparticles and T6 heat treatment on the microstructure and tensile properties of the SiC/AA2024 nanocomposites. In addition, this study also intends to develop a proper processing method which can be used to produce advanced heterostructure AMCs for applications.

## 2. Materials and Methods

Table 1 shows the chemical composition of the raw powder. Two mixtures of gas-atomized AA2024 powder with a composition of Al-4.1Cu-1.6Mg (wt.%) with an average size of 20 μm (as shown in Figure 1a) and SiC nanoparticles with an average size of 50 nm (as shown in Figure 1b) were firstly carried out with planetary ball-milling for 24 h. The process was performed at a speed of 500 rpm with 1 wt.% stearic acid added as a process control agent (PCA) during milling. The diameter of the stainless-steel balls used for the milling was 10 mm, and the ball-to-powder weight ratio was 10:1. According to the literature [4], a nanocomposite with 10 vol.% SiC would display low ductility. As a result, SiC/AA2024 nanocomposite powders with the SiC nanoparticle content of 1% and 5% by volume were produced in the present study. The morphology and particle size distributions of the milled nanocomposite powders are displayed in Figure 1c,d, showing that the mean particle sizes of the 1 and 5 vol.% SiC/AA2024 nanocomposite powders are 22.3 and 18.9 μm, respectively. Figure 2 shows the preparation process of the SiC/AA2024 nanocomposite samples. The as-milled powders were then consolidated via spark plasma sintering (SPS) at 500 °C for 10 min under a pressure of 50 MPa. The as-sintered ingots were induction heated to 500 °C (holding for 5 min) at a rate of 100 °C/min. The hot extrusion process, with an extrusion ratio of 16:1 to produce samples of 7.5 mm in diameter, was carried out in an argon atmosphere. The as-extruded samples were heat-treated (HT) with a condition of solution treatment (502 °C for 2 h), quenching, and aging (140 °C for 4 h). The as-extruded and heat-treated 1 and 5 vol.% SiC/AA2024 nanocomposite samples were named 1SNC and 5SNC, as well as 1SNC-HT and 5SNC-HT, respectively.

The phase structures of the samples were tested using an X-ray diffractometer (XRD) (Rigaku Ultima IV). The particle size measurements were carried out by the Mastersizer 2000 particle size analyzer. To investigate the microstructure evolution and deformation mechanism, scanning electron microscopy (SEM, Tescan GIGA), electron back-scattered diffraction (EBSD, Aztec HKL MAX, Oxford), and transmission electron microscopy (TEM, Talos F200X G2) were utilized. The quasi-static tensile tests were conducted by a Zwick/Roell Z100 testing machine under a strain rate of 5 × 10^−4^/s. Tensile test specimens were cut from the rod samples along the extrusion direction with gauge dimensions of 15 × 3 × 2 mm. Three specimens were tested for each of the rods.

## 3. Results

### 3.1. Morphology of 1 and 5 vol.% SiC/AA2024 Nanocomposites

Figure 3 shows the XRD patterns of the samples, where all the samples exhibited the peaks of Al and SiC. The typical precipitates in the AA2024 alloy: Al_2_Cu (θ) and Al_2_CuMg (S) phases [23], were detected in the 1SNC and 5SNC samples. However, they became weaker or disappeared in 1SNC-HT and 5SNC-HT samples, suggesting that they were possibly dissolved during the solution heat treatment. It is speculated that the θ and S phases would form in the T6 heat-treated samples, but they may be very difficult to detect by XRD due to the tiny sizes and low volume fractions.

Figure 4 demonstrates the back-scattered electron SEM images of the 1SNC, 1SNC-HT, 5SNC, and 5SNC-HT samples, respectively. It can be found that bright Cu-rich precipitate particles were detected in all the samples. Relying on the XRD results and the EDS spectrum, it can be speculated that the bright precipitate is Al_2_CuMg phase [24]. After T6 heat treatment, the precipitates became smaller and more dispersed, as displayed in Figure 4b,d.

Figure 5 presents the EBSD IPF Z images of the microstructures of the 1SNC, 1SNC-HT, 5SNC, and 5SNC-HT samples. High-angle grain boundaries were constructed as the boundaries with a misorientation angle θ equal to or larger than 15°. According to the statistics of EBSD results, the average grain sizes of 1SNC, 1SNC-HT, 5SNC, and 5SNC-HT samples were determined to be 424, 411, 305, and 296 nm, respectively. As shown in Figure 5a,b, bimodal microstructures consisting of coarse elongated grains and fine equiaxed grains were observed in the 1SNC and 1SNC-HT samples, in which the orientations of equiaxed grains were randomly distributed. For the 5SNC sample, a finer microstructure containing ultrafine equiaxed grains and larger elongated grains was observed, as displayed in Figure 5c. The 5SNC-HT sample revealed a similar microstructure, including coarse grains and fine grains after T6 heat treatment. Figure 6 shows the grain size distributions from the statistics of EBSD results. It is noted that due to the low number frequency, the coarse grains were not distinct in the 5SNC and 5SNC-HT samples. This finding also indicated that the counts of coarse grains in 5SNC and 5SNC-HT samples were much less than those in the 1SNC and 1SNC-HT, which means a more pronounced bimodal distribution of grain size in the 1SNC and 1SNC-HT samples than that in the 5SNC and 5SNC-HT samples. Based on the grain size results of EBSD, it can be calculated that the average grain sizes of coarse grain (>1 μm) and fine grain (<1 μm) in the 1SNC and 1SNC-HT samples were 1.41 μm and 360 nm, and 1.52 μm and 346 nm, respectively. For the 5SNC and 5SNC-HT samples, the coarse grain and fine grain were 1.84 μm and 282 nm, and 1.63 μm and 276 nm, respectively. These findings further confirm the presence of a bimodal microstructure in the nanocomposite samples.

The TEM bright-field images of the samples are shown in Figure 7. As displayed in Figure 7a, a large number of coarse lens-shaped precipitates (about 100–300 nm) were observed at the grain boundaries. After T6 heat treatment, the number of coarse precipitates decreased, and the number of fine dispersed precipitates increased in the 1SNC-HT sample. Figure 7c shows a similar but finer microstructure of the 5SNC sample. After T6 heat treatment, more dispersed precipitates were detected, as illustrated in Figure 7d. It is noted that precipitates including nano-scaled or large angular precipitates were observed in all of the heat-treated samples, as demonstrated in Figure 7b,d. Figure 7e depicts a high-magnification TEM image of the microstructure of the 1SNC-HT sample. Based on the EDS point analysis results of the precipitates, it can be concluded that the large precipitates (pointed out by yellow arrows) near the dislocations or at grain boundaries are the typical Al_2_CuMg (S’) precipitates [24,25]. Figure 7f shows the high-resolution TEM image and corresponding Fourier-filtered (FFT) image of nano-scaled round intragranular precipitates in the 1SNC-HT sample (pointed out by red arrows), indicating that the precipitates are likely S’ phase [25] as well. Almost no traditional needle-shaped S’ precipitates were detected in the 1SNC-HT sample. It also can be found that two different kinds of S’ precipitates appeared in the 1SNC-HT sample based on the EDS result and the FFT image. The 5SNC-HT sample showed the same precipitation behavior but more dispersed S’ precipitates.

### 3.2. Tensile Properties

The tensile engineering stress–strain curves of the 1SNC, 1SNC-HT, 5SNC, and 5SNC-HT samples and the design of the tensile testing specimens are shown in Figure 8. The corresponding tensile mechanical properties of the samples are listed in Table 2, including yield strength, ultimate tensile strength, and elongation to fracture. The results indicated that with increasing the volume fraction of SiC nanoparticles (from 1 to 5 vol.%), the as-extruded nanocomposite samples exhibited a clear increase of yield strength (YS) and ultimate tensile strength (UTS), from 413 and 501 MPa to 461 and 565 MPa, respectively, while elongation to fracture exhibited a decreased trend from 5.4% to 4.1%. However, compared with the 5SNC sample, the YS and UTS of the 5SNC-HT sample increased to 565 and 613 MPa, respectively, while the elongation to fracture decreased to 2.4%. Differently, 1SNC and 1SNC-HT samples showed a completely different trend. A simultaneous improvement of strength and ductility was observed in the 1SNC-HT sample, as displayed by the YS, UTS, and elongation at fracture increase from 413 MPa, 501 MPa, and 5.4% to 496 MPa, 572 MPa, and 6.7%, respectively.

Figure 9 shows the tensile fracture surfaces of the SiC/AA2024 nanocomposite samples. The fracture surfaces mainly featured the ductile dimples, tear ridges, and micro-voids [26,27]. The micrographs of the fracture surfaces revealed the mixed ductile and brittle fracture in the 1SNC sample. With increasing the content of SiC nanoparticles from 1 to 5 vol.%, the number of cavities and micro-voids increased, and the sizes of dimples decreased, which may be associated with grain refinement resulting from high nanoparticle content. After T6 heat treatment, the dimples on the fracture surfaces of the 1SNC-HT sample became deeper and larger, indicating the occurrence of ductile fracture. For the 5SNC-HT sample, a different fracture surface morphology was observed. After heat treatment, more micropores [28] with tear ridges were observed in the fracture zone, and ductile dimples were almost negligible, which is consistent with poor ductility.

## 4. Discussion

### 4.1. Microstructures

It is clear that a bimodal microstructure consisting of fine equiaxed grains (hard domains) and elongated coarse grains (soft domains) formed in the extruded 1SNC nanocomposite sample (as illustrated in Figure 5 and Figure 6). The fine equiaxed grains possibly resulted from the dynamic recrystallization, while the elongated coarse grains likely came from the abnormal grain growth during the SPS process. The high grain boundary energy related to the large area of grain boundaries due to the long time spent ball-milling could provide the extra thermodynamic driving force, helping grain growth [20]. After T6 heat treatment, the 1SNC-HT sample maintained a similar microstructure with a slight loss of grain size, from 424 to 411 nm, as displayed in Figure 5a,b and Figure 6a,b. When it comes to the 5SNC sample, a similar microstructure evolution also occurred, only with a much finer grain size of 305 nm compared to the 1SNC sample due to the high addition of SiC nanoparticles (5 vol.%).

Figure 10 shows the STEM bright-field images and the corresponding EDS Si and C mappings of the 1SNC-HT and 5SNC-HT samples. It is noted that because of the low detection sensitivity of C, the distribution of Si element can be treated as the reflection of SiC nanoparticle distribution. As shown in Figure 10b,e, the SiC nanoparticles were mostly distributed at the grain boundaries (pointed out by red arrows in Figure 10a,d). Aggregation of SiC nanoparticles were detected in the 5SNC-HT sample, while they were better dispersed and more uniformly distributed in the 1SNC-HT sample, as shown in Figure 10b,e. Based on the STEM observation results, it can be estimated that the average sizes of SiC nanoparticles were 88 and 110 nm for the 1SNC-HT and 5SNC-HT samples. Due to the dispersion of SiC nanoparticles at grain boundaries, the grain boundary movement was impeded by the Zener pinning effect [29], which consequently inhibited grain growth during heat treatment. As a result, the bimodal microstructure remained, and the average grain size did not change significantly.

As displayed in Figure 7b, after T6 heat treatment, the number of coarse precipitates decreased and amounts of fine dispersed precipitates were observed in the 1SNC-HT sample. A similar microstructure was also found in the 5SNC-HT sample with more and dispersed Sʹ precipitates, as shown in Figure 7d. S’ precipitates of two types appeared in the T6 heat-treated samples. The first type of Sʹ precipitates represents relatively larger lens-shaped particles with sizes of around 50–150 nm (indicated by yellow arrows in Figure 7e), while the second type of S’ precipitates are tiny round precipitates with sizes of 10–20 nm (indicated by red arrows in Figure 7e). The lens-shaped Sʹ precipitates were reported previously [25,30], and they might result from the un-dissolved S phase during solution treatment. The formation of the round-shaped intragranular S’ precipitates is likely related to the amounts of grain boundaries, dislocations, and the addition of SiC nanoparticles, which have been reported to influence the precipitation [2]. The formation of the intragranular Sʹ precipitates is likely associated with the quenched-in vacancies, which could act as nucleation sites for precipitation. It is believed that the presence of amounts of grain boundaries due to grain refinement, high density of dislocations, and the addition of SiC nanoparticles could increase the free energy of the system, which accelerates the precipitation of S’. The accelerated precipitation consumes a large fraction of Cu and Mg solute atoms and then inhibits the growth of the Sʹ nucleated at vacancies, leading to the formation of fine S’ precipitates. For the 5SNC sample, the higher addition of SiC nanoparticles with the high-energy ball-milling process can generate more dislocations, which helps with nucleation, resulting in more and dispersed Sʹ phases. The intragranular precipitates can effectively impede the dislocation movement during plastic deformation, which contributes to the improved work-hardening ability.

### 4.2. Mechanical Properties

As shown in Figure 3, we can roughly consider that there are only four phases (Al, SiC, S, and θ) in SiC/AA2024 nanocomposites. To further study this, the relative content of phases can be calculated according to the RIR method in JADE from the XRD results. Based on this assumption, the calculated amounts of Al_2_Cu (θ) were 1.6, 1.2, 2.1, and 1.5 wt.% and the amounts of Al_2_CuMg (S) were 1.3, 1.8, 1.4, and 1.1 wt.% in the 1SNC, 1SNC-HT, 5SNC, and 5SNC-HT samples, respectively. Along with the increase of SiC content, the amounts of S and θ increased because of more nucleation sites. After HT, the amounts of S and θ decreased, which may be attributed to the dispersed and smaller precipitates.

The tensile test results in Figure 8 indicate that with increasing the SiC nanoparticle content from 1 to 5 vol.%, the YS and UTS increased from 413 and 501 MPa to 461 and 565 MPa, respectively, and the elongation to fracture decreased from 5.4% to 4.1%. The addition of SiC nanoparticles led to a linear increase of the composites as long as the nanoparticles were uniformly distributed in the matrix. The increase may be attributed to the stronger grain boundary strengthening (finer grain size shown in Figure 5) and nanoparticle strengthening (more SiC shown in Figure 10) of the 5SNC sample. In this sense, SiC nanoparticles can act in both direct and indirect roles in strengthening nanocomposites. However, as the content of SiC nanoparticles increases, it is inevitable to form the SiC clusters in the present composites, which affects the El. On one hand, voids might exist in clusters, which act as microcrack sources. On the other hand, the clusters of reinforcement induced stress concentration at the interface and accelerated the propagation of cracks [31]. 

For the 1SNC-HT sample, a simultaneous improvement of strength and tensile ductility occurred when compared to the 1SNC sample. This may result from the following. The first is the ability of producing and accumulating geometrically necessary dislocations (GNDs) of the samples. The bimodal microstructure consisting of ultrafine grains (UFGs) and coarse grains (CGs) was detected in the 1SNC and 1SNC-HT samples, as shown in Figure 5 and Figure 6. During tensile deformation of the bimodal materials, plastic deformation occurred first in the soft domains (CGs), while the hard domains (UFGs) still remained elastic. Therefore, the soft domains cannot undergo plastic deformation due to the constraint of the hard domains. Since the strain must be continuous across the interface, the soft domains need to produce strain gradients [32]. Then, GNDs in CGs will be produced and piled up against domain boundaries to accommodate strain incompatibility, resulting in long-range internal stress, such as back stress [16]. The back stress is directional and counteracts some of the applied shear stress, making the soft domain stronger to withstand higher applied stress [19].

The distribution of GNDs in 1SNC samples can be described by means of the local misorientation calculated by EBSD, as shown in Figure 11a–d. After the tensile test, the GNDs were primarily accumulated, as indicated by the darker color in Figure 10d and Figure 11b. The calculated dislocation densities of the 1SNC and 1SNC-HT samples were 3.11 × 10^15^ and 2.57 × 10^15^, respectively. After tensile deformation, the dislocation densities increased to 4.33 × 10^15^ and 5.82 × 10^15^, respectively. The increment of dislocation density in the 1SNC-HT sample was more significant than that in the 1SNC sample during tensile testing, which means an increased ability of accumulating GNDs in the 1SNC-HT sample. This might be due to the low dislocation density in the 1SNC-HT sample after HT, which resulted in larger strain gradients between the soft domains (CGs) and the hard domains (UFGs) to help with producing more back stress. Combining the EBSD KAM map and dislocation density, it is quantitatively speculated that the 1SNC-HT sample can produce and accumulate an increased number of GNDs during tensile deformation, which is advantageous for an increase of ductility.

Another reason for the enhancement of both strength and ductility after heat treatment may be related to the fine intragranular precipitates. In UFG materials, after releasing from GBs, the dislocations move to the opposite grain boundary and are annihilated [33]. This leads to a remarkable decrease in work-hardening ability. In this study, the formation of nano-scaled precipitate Sʹ in the grains of the heat-treated sample helped to accommodate dislocations by retarding dislocation movement more effectively during tensile deformation. As shown in Figure 12a, the dislocations (pointed out by yellow arrows) were hindered and impeded by fine intragranular precipitates (pointed out by red arrows) in the deformed sample. This effect might inhibit the annihilation of dislocations at GBs [34], making a positive contribution to the improvement of strength and ductility. As a result, the 1SNC-HT sample revealed an improvement in both strength and ductility, with the YS, UTS, and El increasing from 413 MPa, 501 MPa, and 5.4% to 496 MPa, 572 MPa, and 6.7%, respectively.

In contrast, after heat treatment, the YS and UTS of the 5SNC-HT sample increased from 461 and 565 MPa to 594 and 613 MPa, respectively, while the elongation at fracture declined remarkably from 4.1% to 2.4%. On one hand, the concentration of SiC nanoparticles in the 5SNC-HT sample was much higher than that in the 1SNC-HT sample. It is believed that cavities first form between SiC nanoparticles and the Al matrix at a large tensile strain during tensile testing [4], so that the cavities grow and coalesce to cause premature failure of the composite material, as shown in Figure 9d. On the other hand, the smaller grain size and more nanoparticles led to a weak strain-hardening ability [35]. Besides, the obvious agglomeration of SiC nanoparticles in the 5SNC-HT sample was also observed in Figure 9e, which is harmful to the ductility of nanocomposite. To sum up, the finer grain size along with more concentrated SiC nanoparticles caused the sharp ductility decrease of 5SNC-HT.

## 5. Conclusions

SiC/AA2024 nanocomposites with 1 and 5 vol.% SiC nanoparticles were prepared with the powder metallurgy method, including high-energy ball-milling, spark plasma sintering, and hot extrusion, followed by T6 heat treatment. The microstructure evolution, precipitation behavior, and mechanical properties were investigated. The following conclusions were drawn:

(1)A bimodal microstructure formed in the as-extruded samples and still remained after T6 heat treatment, which is likely owing to the pinning effect of the dispersed SiC nanoparticles on grain boundaries.(2)With increasing SiC nanoparticle content from 1 to 5 vol.%, the yield strength and ultimate tensile strength increased from 413 and 501 MPa to 461 and 565 MPa, respectively, and the elongation to fracture decreased from 5.4% to 4.1%.(3)After T6 heat treatment, the 1 vol.% SiC/AA2024 nanocomposite sample revealed an improvement in both strength and ductility, where the YS, UTS, and El increased from 413 MPa, 501 MPa, and 5.4% to 496 MPa, 572 MPa, and 6.7%, respectively.(4)The simultaneous increase in strength and ductility in the T6 heat-treated 1 vol.% SiC/AA2024 nanocomposite sample may be attributed to the bimodal microstructure, which could generate back stress strengthening during tensile deformation, as well as the dispersed intragranular Sʹ precipitates which could accumulate dislocations by blocking dislocation movement.

## Figures and Tables

**Figure 1 materials-15-03547-f001:**
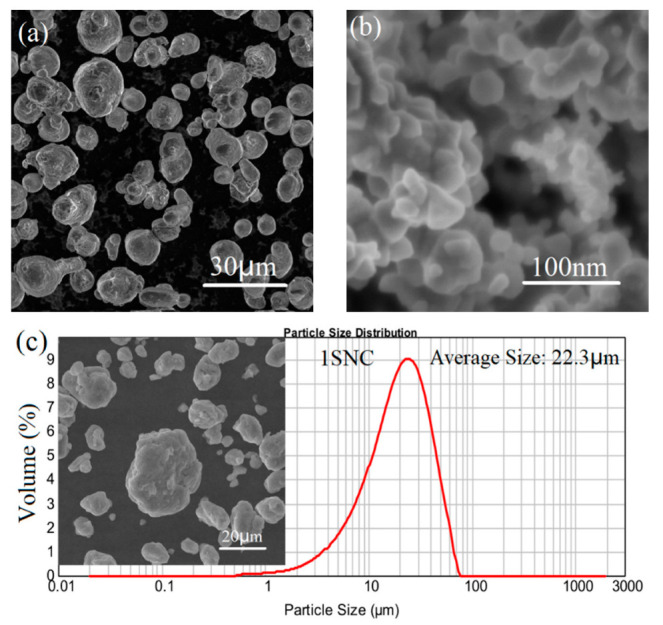
Particle morphology of raw powders: (**a**) gas-atomized AA2024 powder particles, (**b**) SiC nanoparticles, and (**c**,**d**) particle morphology and size distributions of the milled 1 and 5 vol.% SiC/AA2024 nanocomposite powders.

**Figure 2 materials-15-03547-f002:**
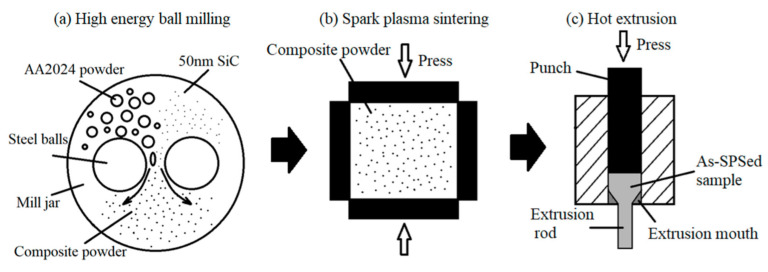
Schematic drawings illustrating the process used to prepare the samples.

**Figure 3 materials-15-03547-f003:**
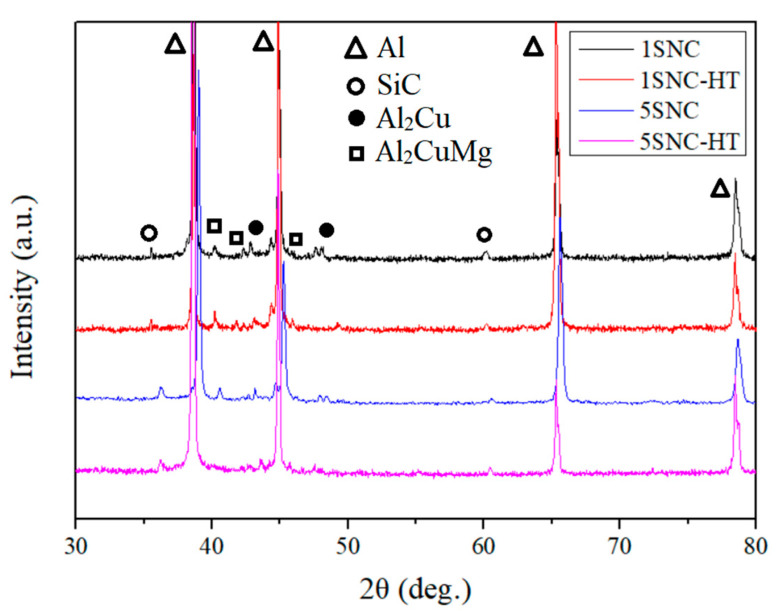
XRD patterns of the 1SNC, 1SNC-HT, 5SNC, and 5SNC-HT samples.

**Figure 4 materials-15-03547-f004:**
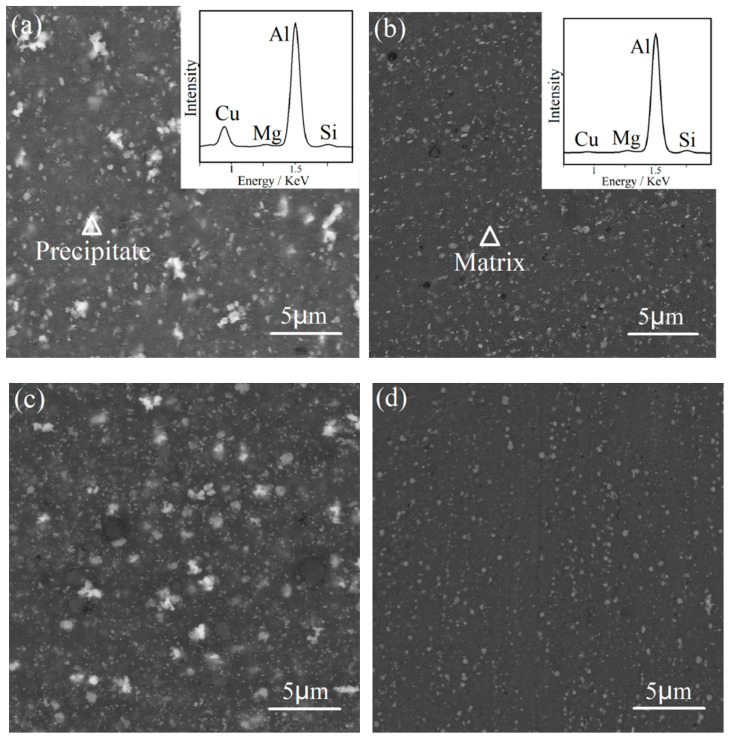
Back-scattered electron SEM images of the (**a**) 1SNC, (**b**) 1SNC-HT, (**c**) 5SNC, and (**d**) 5SNC-HT samples. The EDS spectrums of the precipitate and matrix are shown in the insets, indicated by the triangle mark in (**a**,**b**).

**Figure 5 materials-15-03547-f005:**
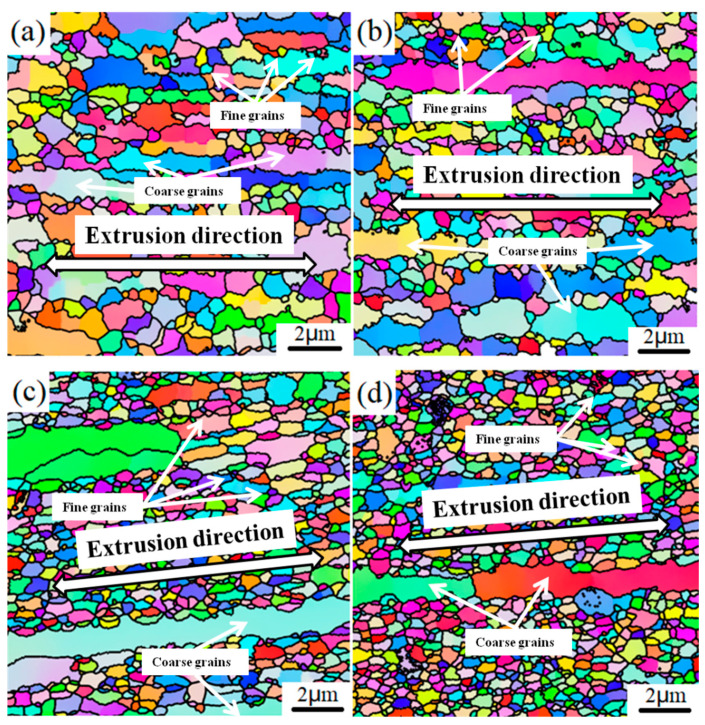
EBSD IPF Z images of the longitudinal sections of (**a**) 1SNC, (**b**) 1SNC-HT, (**c**) 5SNC, and (**d**) 5SNC-HT samples.

**Figure 6 materials-15-03547-f006:**
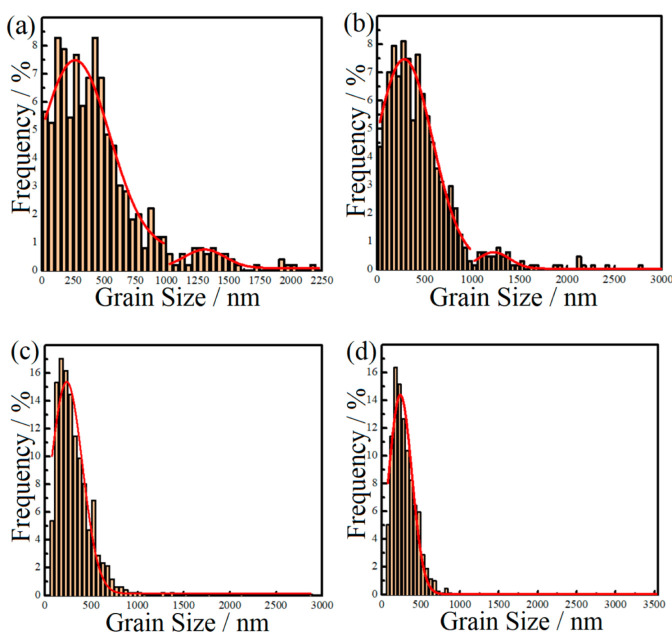
Aluminum grain size distributions of the nanocomposite samples: (**a**) 1SNC, (**b**) 1SNC-HT, (**c**) 5SNC, and (**d**) 5SNC-HT.

**Figure 7 materials-15-03547-f007:**
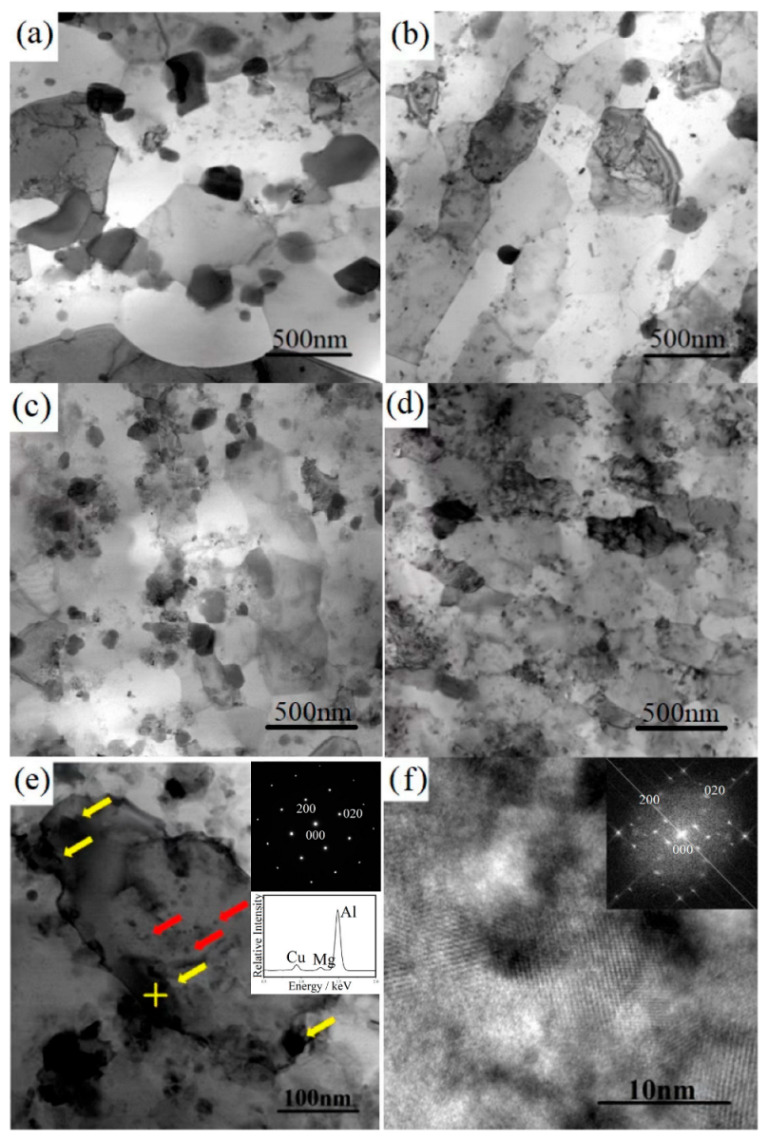
TEM bright-field images of the samples: (**a**) 1SNC, (**b**) 1SNC-HT, (**c**) 5SNC, and (**d**) 5SNC-HT. (**e**) High-magnification TEM image of the 1SNC-HT sample and EDS result of the zone marked by the yellow cross. (**f**) HRTEM image of the precipitate in (**e**), the corresponding FFT pattern is shown in the inset.

**Figure 8 materials-15-03547-f008:**
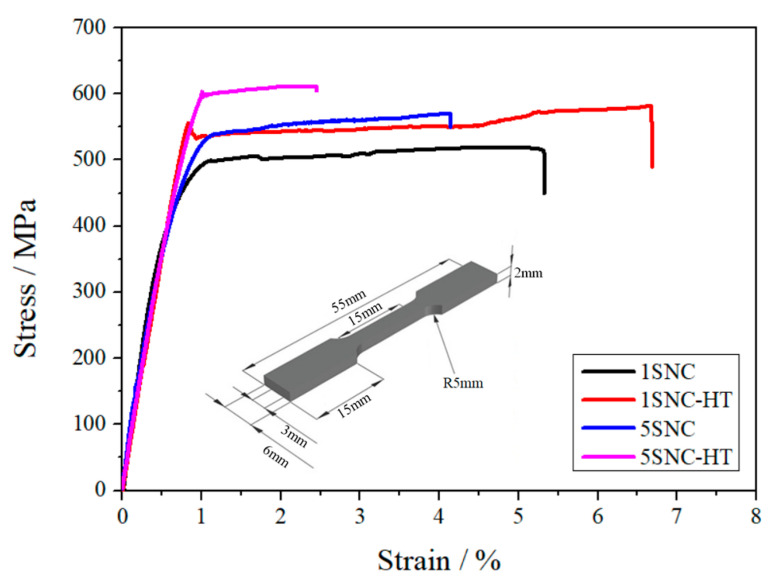
Tensile engineering stress–strain graphs of the 1SNC, 1SNC-HT, 5SNC, and 5SNC-HT samples. The inset is the sketch drawing of the tensile testing specimens.

**Figure 9 materials-15-03547-f009:**
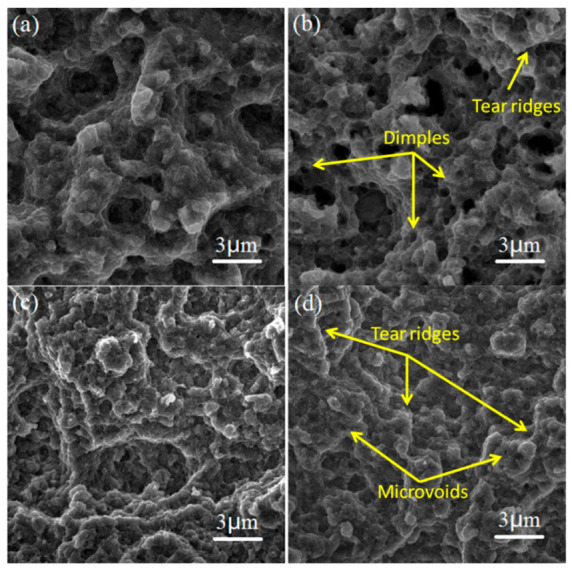
SEM secondary electron images of the fracture surfaces of the tensile test specimens cut from the SiC/AA2024 nanocomposite samples: (**a**) 1SNC, (**b**) 5SNC, (**c**) 1SNC-HT, and (**d**) 5SNC-HT.

**Figure 10 materials-15-03547-f010:**
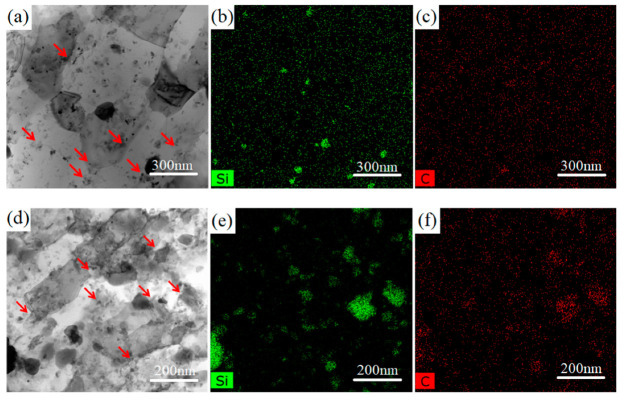
(**a**) STEM bright-field image and (**b**,**c**) corresponding EDS Si and C mappings of 1SNC-HT. (**d**) STEM bright-field image and (**e**,**f**) corresponding EDS Si and C mappings of 5SNC-HT.

**Figure 11 materials-15-03547-f011:**
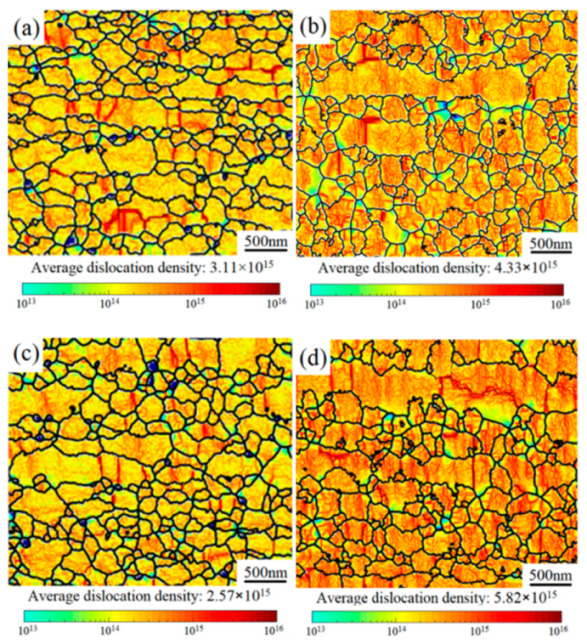
EBSD KAM maps showing the GNDs distribution: (**a**) 1SNC, (**b**) 1SNC after tensile test, (**c**) 1SNC-HT, and (**d**) 1SNC-HT after tensile test.

**Figure 12 materials-15-03547-f012:**
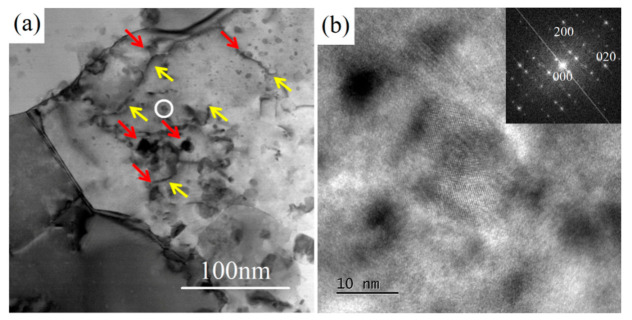
(**a**) TEM bright-field images of the 1SNC-HT sample after the tensile test. (**b**) HRTEM image of the precipitate circled in (**a**), the inset is the corresponding FFT pattern.

**Table 1 materials-15-03547-t001:** Chemical composition of the AA2024 powder (wt. %).

Element	Cu	Mg	Fe	Mn	Si	Zn	Al
Composition	4.15	1.56	0.22	0.75	0.16	0.16	Bal.

**Table 2 materials-15-03547-t002:** Tensile mechanical properties of different SiC/AA2024 nanocomposite samples.

Samples	Yield Strength/MPa	Ultimate Tensile Strength/MPa	Elongation to Fracture/%
1SNC	413 ± 5	501 ± 8	5.4 ± 0.4
1SNC-HT	496 ± 9	572 ± 12	6.7 ± 0.6
5SNC	461 ± 7	565 ± 9	4.1 ± 0.3
5SNC-HT	594 ± 10	613 ± 17	2.4 ± 0.8

## Data Availability

Not applicable.

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
