# Peer review of "Investigation of Microstructures and Mechanical Properties of SiC/AA2024 Nanocomposites Processed by Powder Metallurgy and T6 Heat Treatment"

_materials, 2022, doi:10.3390/ma15103547_

Round 1

Reviewer 1 Report

1)The introduction section needs a substantial addition. A large number of studies have been carried out in this area. Please describe in more detail how yours differs from previous work in this area.

2) The abbreviation CNT is not decrypted.

3) The standard for mechanical testing and samples should be provided. 

4) The indexes and sone axis should be given in the FFT Fig. 6e,f

5) The error bars should be provided for the YS, UTS, El. values

Reviewer 2 Report

This article deals with an experimental study on the correlations between microstructure and tensile properties of SiC/AA2024 nanocomposites processed by powder metallurgy and hot extrusion.  Specimens with nanoparticle contents of 1 and 5% by volume were produced. The effects of T6 heat treatment on these specimens were also investigated and discussed, showing advantages on both mechanical strength and ductility.  

The paper is interesting and complete in the treatment of the set objectives: the abstract is concise and supported by the results; introduction provides an exhaustive background; the paragraphs on methodology and results are well organized; discussion and conclusions are in line with the objectives.

In my opinion, it is a really good work and deserves to be published after only a minor revision of the English language.

Line     

38        Correct “…caused the elongation to fracture to decrease...” in “…caused fracture elongation to decrease...”

39        Does CNT stand for carbon nanotube? Please, explain this acronym.

69        Correct “…contents of 1 and 5 vol.%...” in “…contents of 1 and 5 % by volume...”

212      Correct “… which is in keeping with the poor ductility.” in “which is consistent with poor ductility.”

273      Correct “… can serve as the nucleation sites…” in “can act as nucleation sites…”

277      Correct “…can bring in more dislocations, …” in “can generate more dislocations, …”

296      Correct “…the constraint from the hard domains…” in “…the constraint of the hard domains…”

321      Correct “…which is benefit to the increase of ductility.” in “…which is advantageous for an increase in ductility.”

330      Correct “… decrease of…” in “…decrease in…”

346      Correct “…, such that…” in “…, so that…”

371      Correct “…elongation to fracture…” in “…elongation at fracture…” or “…fracture elongation…”

374      Correct “…increase of strength and ductility…” in “…increase in strength and ductility…”

Reviewer 3 Report

Review report on the topic ‘SiC/AA2024 nanocomposites processed by powder metallurgy and T6 heat treatment: A study of microstructures and mechanical properties’. Comments are listed below:

  1. Strengthen the abstract section. Add the key conclusion of the works in the last two lines of the abstract section. Also, remove the unnecessary information.
  2. There are numerous spelling and grammatical errors. Please revise the manuscript thoroughly. Sentences are also not complete and references are also cited in a rough manner. Eg. ‘After T6 heat treatment, a simultaneously improvement of the strength and 18 ductility (YS: 413 → 496 MPa, UTS: 501 → 572 MPa, EL: 5.4% → 6.7%) was observed in the 19 1vol.%SiC/AA2024 nanocomposite sample’.
  3. Discuss the novelty of the work in respect of the application.
  4. Try to make a bridge between current and previously published work and specify the gap area and objective of the work. Also, discuss the major process used for such type of deposition and the problem associated with it. Discuss the advantage of the processing route, i.e. powder metallurgy.
  5. Discuss the selection of SiC weight percentage.
  6. Provide the image of the experimental setup with good quality.
  7. How was the particle size measured?
  8. Provide the relative amount of the phases using the XRD results.
  9. Provide a good quality image (Fig. 3).
  10. EBSD map needs more discussion. In present, it looks like a technical report.
  11. Provide the image of the standard tensile specimen and fractured specimen.
  12. The fracture surface study needs a clear discussion. Refer to the following: https://doi.org/10.1016/j.engfailanal.2016.06.012; https://doi.org/10.1016/j.engfailanal.2017.06.044.
  13. How the increase in weight percentage of SiC affects the mechanical properties? Try to relate the variation in mechanical properties with microstructure.
  14. Shorten the length of the conclusion section. Keep only key points.

Reviewer 4 Report

Dear Authors and Editors

The article carries hardly any scientific soundness in interest with materials reader. The article doesnot contain no quality data presentation overall.  Because the title is too long. Heat treatment is part of the process for composite to enhance microstructure and mechanical strength. So in this point i donot find anything new in this article. Authors claim bimodal distribution. Hardly it could able to see any bimodal particle either in morphology or in particle distribution in graphs.

such as

Bimodal distribution of filler on viscosity and thermal expansion of glass composites, Ceramics International, Volume 39, Issue 2, March 2013, Pages 1659-1666.

The effect of heat treatment on titania slag, Journal of Minerals and Materials Characterization and Engineering 9 (09), 795

What is the main question addressed by the research?

Is it relevant and interesting? It does not sound either relevant or interesting.

How original is the topic?

The topic looks less original, as it adhere the standard procedure of heating for development of microstructutre.

What does it add to the subject area compared with other published material?

Authors need to reply this.

Is the paper well written?

Yes, the authors organized the paper well. However there are some images out of focus.

Are the conclusions consistent with the evidence and  arguments presented?

There are some facts which does not meet the claim authors made such as bimodal.  

Do they address the main question posed?

Whats the main question in this article? How authors approach this?

Round 2

Reviewer 1 Report

Thanks to the authors for correcting the article.

1) as a small remark, I would like to add to the introduction an analysis of the influence of other materials, for example B4C
2) in Figure 12 it is necessary to designate the indexes and the axis of the zone.

Reviewer 3 Report

Accept in present form.

Author Response

Comments and Suggestions for Authors: Accept in present form.  

Response:

Thank you very much for your kindness.

Reviewer 4 Report

Dear Authors

The article has been improved from the previous version, however there are some related articles need to concern which use powder metallurgical route for preparation of composite and need to include such as 

Net-Shape NiTi Shape Memory Alloy by Spark Plasma Sintering Method

Appl. Sci. 11 (4), 1802,2021   Fabrication of thermal plasma sprayed NiTi coatings possessing functional properties Coatings 11 (5), 610, 2021   Also its important to show the authors coarse and fine grain distribution in composite to support bimodal terminology.    Do the authors claim T6 heat treatment is new in this field?    Author need to clarify "unusual round-shaped precipitate S’ comes from the mounts of grain boundaries, dislocations and the addition of SiC nanoparticles" whats this composition ? What this shape signify? Where it comes form? I donot see this S form in the article.      our investigation on the AA2024 nanocomposites could adjust the processing parameters to obtain the optimized strength and ductility   Author need to specify the sentence above.    However the article has been improved, still some of the claims has not support in the manuscript. Authors need to work on that. 
